# Targeting of Phospholipase D1 Ameliorates Collagen-Induced Arthritis via Modulation of Treg and Th17 Cell Imbalance and Suppression of Osteoclastogenesis

**DOI:** 10.3390/ijms21093230

**Published:** 2020-05-02

**Authors:** Hyun Jung Yoo, Won Chan Hwang, Do Sik Min

**Affiliations:** 1Department of Molecular Biology, College of Natural Science, Pusan National University, Busan 46241, Korea; totoro77@snu.ac.kr (H.J.Y.); deielia@daum.net (W.C.H.); 2College of Pharmacy, Yonsei University, Incheon 21983, Korea

**Keywords:** phospholipase D1, rheumatoid arthritis, regulatory T cell, Th17 cell, osteoclastogenesis

## Abstract

Phospholipase D1 (PLD1) plays a crucial role in various inflammatory and autoimmune diseases. Rheumatoid arthritis (RA) is a chronic and systemic autoimmune disease. However, the role of PLD1 in the pathogenesis of RA remains unknown. Here, we first investigated the role and effects of PLD1 in collagen-induced arthritis (CIA) and found that genetic and pharmacological inhibition of PLD1 in DBA1/J mice with CIA reduced the incidence of CIA, decreased the clinical score, and abrogated disease symptoms including infiltration of leukocytes, synovial inflammation, bone erosion, and cartilage destruction. Moreover, ablation and inhibition of PLD1 suppressed the production of type II collagen-specific IgG2a autoantibody and proinflammatory cytokines, accompanied by an increase in the regulatory T (Treg) cell population and a decrease in the Th17 cell population in CIA mice. The PLD1 inhibitor also promoted differentiation of Treg cells and suppressed differentiation of Th17 cells in vitro. Furthermore, the PLD1 inhibitor attenuated pathologic bone destruction in CIA mice by suppressing osteoclastogenesis and bone resorption. Thus, our findings indicate that the targeting of PLD1 can ameliorate CIA by modulating the imbalance of Treg and Th17 cells and suppressing osteoclastogenesis, which might be a novel strategy to treat autoimmune diseases, such as RA.

## 1. Introduction

Rheumatoid arthritis (RA) is a chronic, systemic autoimmune inflammatory disorder that progressively destroys the articular cartilage and bones of the joints [1]. Although both environmental and genetic factors are involved in the etiology of RA, the precise mechanism of onset remains unclear. Elucidation of the mechanism of bone destruction associated with RA can contribute to understanding the mode of action and establishment of the therapeutic strategies for RA [2]. Bone resorption by osteoclasts in the synovium is implicated in bone damage in RA [3]. RA synovial fibroblasts (RASF) play a pivotal role in the development of inflammation and joint destruction. RASF have been described as transformed cells, which share morphologic features, such as resistance to apoptosis, with tumor cells. [4,5]. Phospholipase D (PLD) is considered a promising target in treating inflammation and cancer [6,7,8]. PLD catalyzes the hydrolysis of phospholipids to phosphatidic acid (PA)—a bioactive molecule implicated in a variety of cellular processes, such as cell proliferation, differentiation, migration, and membrane trafficking [9]. Two isoforms of phosphatidylcholine-specific PLD, PLD1, and PLD2, have been identified and characterized [10,11]. Notably, the expression of PLD1 but not PLD2, is upregulated in various inflammatory and autoimmune diseases, including acute pancreatitis [12], peritonitis [13], RA [14], and Alzheimer disease [15]. PLD1, and not PLD2, is a transcriptional target of proinflammatory cytokines. PLD1 plays a pivotal role in synoviocyte activation and is tightly linked to the production of various cytokines/chemokines to form an interactive molecular network for the perpetuation of RA [14]. PLD2 expression is not affected in activated monocytes, and it appears to be a constitutive enzyme in circulating monocytes [16]. PLD1 expression correlates positively with the severity of RA; thus, abnormal upregulation of PLD1 might contribute to the pathogenesis of chronic arthritis [14]. However, the functional role of PLD1 in RA pathogenesis remains poorly defined. T cell-mediated autoimmune responses, where IL-17-producing T helper (Th) cells act as crucial effectors, play a critical role in RA pathogenesis. IL-17 plays an important role in the development of collagen-induced arthritis (CIA) in the CIA mouse model, a widely used animal model, by activating autoantigen-specific immune responses. Regulatory T (Treg) cells are crucial players in the prevention of autoimmunity and impaired Treg cell function is associated with RA. Thus, the Treg/Th17 imbalance is a critical determinant of the pathogenesis of arthritis. PLD1 has been reported to play an important role in Treg differentiation and T-cell activation [17,18,19,20]. However, it remains unknown whether targeting of PLD1 could affect the pathogenesis of RA via modulation of Treg/Th17 imbalance in the CIA model. In this study, we investigated the effects and molecular mechanisms of PLD1 in CIA using PLD1 knockout mice and the PLD1-selective inhibitor, VU0155069 [21].

## 2. Results

### 2.1. Deficiency of PLD1 Results in Reduced CIA Severity

We investigated whether PLD1 could play a role in CIA, using the CIA mouse that mimics many features of human RA. We backcrossed *PLD1*^−/^^−^ C57BL/6 mice to the DBA/1J strain, in which CIA is the most consistently seen [18], and immunized them with collagen. Photographs of hind paws showed great reduction in inflammation in *PLD1*^−/^^−^ mice compared to that of their wild-type (WT) littermates (*PLD1*^+/+^) (Appendix A). As analyzed by the arthritis score and incidence, *PLD1*^+/+^ DBA1/J mice developed signs of arthritis following immunization (Figure 1A,B). However, the joints of CIA-challenged *PLD1*^−/^^−^ mice showed reduced progression of CIA compared with that in the joints of *PLD1*^+/+^ mice (Figure 1A,B), suggesting that PLD1 promotes disease severity in arthritis. Histological analysis of hematoxylin and eosin (H&E) stained knee joint sections revealed that CIA-challenged *PLD1*^−/^^−^ mice showed significant suppression of the symptoms, including decrease in infiltration of leukocytes, synovial inflammation, bone erosion, and cartilage destruction (Figure 1C,D). We further evaluated bone erosion in naïve (normal) and arthritic mice with the use of micro-computerized tomography (CT) (Figure 1E). WT littermate mice with CIA exhibited profound osteolytic lesions with markedly increased erosion of the inflamed femur (metaphysis) and knee joint (Figure 1E). On the contrary, *PLD1*^−/^^−^ mice recovered remarkably from bone erosion induced by CIA (Figure 1E). The tibial trabecular bone volume [%, BV/tissue volume (TV)] was significantly reduced in *PLD1*^+/+^ mice with CIA compared to that in non-arthritic mice (Figure 1F). The reduction in trabecular bone mass in *PLD1*^+/+^ mice was largely owing to a reduction in trabecular thickness (Tb.Th) and trabecular number (Tb.N), and an increase in trabecular separation (Tb.Sp) (Figure 1F). The loss of trabecular bone parameters was significantly prevented in *PLD1*^−/^^−^ mice. Additionally, trabecular separation was markedly lower in *PLD1*^−/^^−^ mice than in their WT littermates (Figure 1F), suggesting a higher volume and quality of preserved trabecular bone in *PLD1*-deficient mice. Overall, these results indicate that *PLD1* deficiency not only inhibits inflammatory arthritis but also effectively ameliorates joint damage and protects against subsequent bone loss.

### 2.2. PLD1 Ablation Suppresses Collagen Type II-Specific Humoral Response and Production of Proinflammatory Cytokines in CIA Mice

CIA is triggered by host immune responses to type II collagen (CII). Antibody response is of central importance because B cell-deficient mice do not develop the disease, whereas transfer of monoclonal antibodies against CII can induce full-blown arthritis.

To determine the effect of PLD1 on the humoral anti-collagen response, we measured the levels of anti-CII autoantibody and its subclasses in the serum. Serum concentration of anti-CII total IgG was significantly decreased in *PLD1*-deficient mice than in *PLD1*^+/+^ mice (Figure 2A). The levels of anti-CII IgG1 and IgG2a serve as valuable in vivo markers of the TH2 and Th1 response, respectively [22]. No significant differences in the levels of anti-IgG1 were observed between *PLD1*^+/+^ and *PLD1*^−/^^−^ mice (Figure 2A). However, the amount of anti-CII IgG2a antibody was significantly reduced in *PLD1*^−/^^−^ mice compared with that in *PLD1*^+/+^ mice (Figure 2A). Anti-CII IgG2a is associated with cartilage binding and complement fixation [22]. This result suggests that *PLD1* deficiency suppresses the generation of pathogenic autoantibodies to CII, especially anti-CII IgG2a expressed by B cells during CIA. The pathogenic events that lead to the development of human RA are not fully understood, although the pivotal role of proinflammatory cytokines in the induction and maintenance of RA is well documented [23]. Thus, we investigated whether PLD1 could modulate the inflammatory process by regulating cytokine secretion. Although, compared to naïve mice, *PLD1*^+/+^ mice with CIA showed a substantial increase in the levels of proinflammatory cytokines, *PLD1* ablation significantly suppressed the production of proinflammatory cytokines, including TNF-α, IL-6, IL-17, IL-1β, and IFN-γ, compared with the proinflammatory cytokine levels in *PLD*1^+/+^ mice (*p* < 0.01). Several biologic agents are widely used to relieve the symptoms of RA; notable examples of these include anti-TNF agents. *PLD1*-deficient mice showed a decrease in the proportion of TNF-α positive cells in the synovial tissues subjected to CIA, as seen by immunofluorescence staining (Appendix A). These results suggested that *PLD1* deficiency suppresses collagen type II-specific humoral response and production of proinflammatory cytokines in CIA mice.

### 2.3. PLD1 Ablation Regulates the Population of Th17 and Treg Cells

Imbalance of Th17/Treg cells plays a pivotal role in RA pathology. To examine whether the Th17 and Treg cell population was altered in *PLD1*-ablated CIA mice, the ratio of CD4^+^IL-17^+^ (Th17 cells), CD4^+^IFN-γ^+^ (Th1 cells), and CD4^+^Foxp3^+^ cells (Treg), in CD4^+^ T cells of the spleen was assessed by flow cytometry. Compared to arthritic control mice, in DBA1/J mice with CIA, *PLD1* deficiency reduced the population of Th1 and Th17 cells in the spleen (Figure 3A,B). However, the population of Treg cells (CD25^+^Foxp3^+^) in CD4^+^ T cells was dramatically increased in the spleen of CIA-induced PLD1^−/−^ mice compared with the population of Treg cells in the WT littermate mice (4.03% versus 36.7%) (Figure 3C). In this study, we analyzed Treg cells using Foxp3, a typical marker for Treg cells, although there are other Treg markers such as CTLA4 and GARP. Moreover, *PLD1* deficiency showed an increased proportion of CD25^+^Foxp3^+^ cells in the spleen of mice subjected to CIA, as seen by immunofluorescence staining (Figure 3D). These results suggested that *PLD1* deficiency decreased the population of inflammatory Th1 and T17 cells and increased the population of anti-inflammatory CD25^+^Foxp3^+^ Treg cells in vivo.

### 2.4. Pharmacological Inhibition of PLD1 Ameliorates Arthritis and Bone Erosion in CIA Mice

We further investigated whether pharmacological inhibition of PLD1 could affect the severity of CIA. CIA mice were treated with an intraperitoneal injection of the PLD1 inhibitor (VU0155069, 5 mg/kg) or vehicle. The PLD1 inhibitor delayed the onset of arthritis and decreased the incidence of arthritis, arthritic score, and the number of affected limbs (Figure 4A–C). Photographs of hind paws showed that the PLD1 inhibitor greatly reduced inflammation of the hind paws compared to that after vehicle treatment (Figure 4D). We further examined changes in bone microarchitecture in the joint and paws. As assessed by micro-CT, compared to the vehicle, the PLD1 inhibitor markedly suppressed severe bone erosion in the femur, knee joint, and hind paw joints of CIA mice (Figure 4E). We further assessed the trabecular structure of the tibiae by micro-CT and 3D reconstruction. The loss of trabecular bone parameters, including BV, trabecular thickness, and trabecular number was significantly inhibited by treatment with the PLD1 inhibitor (Figure 4F). In addition, the trabecular separation in the PLD1 inhibitor-treated CIA mice was significantly lower than in the vehicle-treated mice (Figure 4F). These data suggested a significant protective effect of the PLD1 inhibitor against joint pathology and local bone loss in CIA mice. Moreover, histopathological assessment of the knee joint section revealed that treatment with the PLD1 inhibitor showed only a few pathological changes and reduced histological scores, including synovial inflammation, bone erosion, cartilage damage, and leukocyte infiltration, compared to those in the vehicle groups (Figure 4G,H). Additionally, the size of the spleen from the vehicle-treated mice with CIA significantly increased compared with that from the non-arthritic mice, but the PLD1 inhibitor markedly recovered the size of the spleen increased by CIA. H&E staining of sections from the liver and spleen showed that, compared with vehicle treatment, PLD1 inhibitor treatment decreased the infiltration of inflammatory cells (Appendix A). Collectively, these data suggested that pharmacological inhibition of PLD1 reduced the arthritic score and inflammation, and prevented damage to bone and cartilage tissues in knee joints of CIA mice.

### 2.5. PLD1 Inhibition Suppresses the Production of Anti-CII IgG2a and Proinflammatory Cytokines in CIA Mice

To address the potential immunogenicity of the PLD1 inhibitor, we measured specific antibody responses against CII in the serum. The titers of Th2-dependent CII-specific IgG1 antibodies increased in PLD1 inhibitor-treated mice compared with those in vehicle-treated mice (Figure 5A). In contrast, PLD1 inhibition significantly reduced the production of anti-CII IgG2a antibody compared with that after vehicle treatment (Figure 5A). These results suggested an important role for the PLD1 inhibitor in modulating the B-cell response, as well as the Th1/Th2 balance in the immune response. Proinflammatory cytokines promote the deleterious imbalance in bone metabolism and contribute to enhanced bone destruction [24]. The PLD1 inhibitor significantly decreased the production of proinflammatory cytokines and chemokines in the serum of CIA mice (Figure 5B). The PLD1 inhibitor also decreased the expression of proinflammatory cytokines induced by CIA, as visualized by immunohistochemistry of knee joint tissues (Appendix A). In addition, the PLD1 inhibitor markedly decreased the production of TNF-α or LPS-induced proinflammatory cytokines (TNF-α, IL-1β, and IL-17) in RASF cocultured with peripheral blood mononuclear cells (PBMC) activated with anti-CD3 and CD28 antibodies (Appendix A). Moreover, the PLD1 inhibitor increased the production of IL-10, an anti-inflammatory cytokine, which had been suppressed by TNF-α or LPS (Appendix A). These results suggested that PLD1 inhibition can inhibit the production of proinflammatory cytokines in arthritic mice and RASF cocultured with PBMC, in addition to that of anti-collagen II IgG2a.

### 2.6. Targeting PLD1 Increases the Population of Foxp3^+^ Treg Cells, Reduces the Population of Th17 Cells and Proliferation of Effector T Cells

We further investigated whether PLD1 inhibition could affect the population of Th17 and Treg cells in CIA mice. Compared to non-arthritic control mice, CIA mice showed an increased population of CD4^+^IL-17^+^ (Th17) cells and reduced population of CD4^+^Foxp3^+^ (Treg) cells (Figure 6A,B). This arthritis-associated Th17/Treg cell imbalance was rescued by the PLD1 inhibitor. Compared to vehicle-treated CIA mice, PLD1 inhibitor-treated CIA mice displayed a decrease in Th17 cell population and an increase in Treg cell population (Figure 6A,B). We further investigated whether the PLD1 inhibitor could affect the population of Treg and Th17 cells differentiating from splenic CD4^+^ T cells. Compared to vehicle treatment, PLD1 inhibitor treatment significantly increased the population of CD4^+^CD25^+^Foxp3^+^Treg cells (52.9% versus 92.8%) and markedly reduced the population of CD25^+^IL-17^+^ Th17 cells (88.2% versus 49.9%) (Figure 6C,D). Forward scatter, side scatter, and gating strategy for the staining of intracellular Foxp3 and IL-17 are shown in Appendix A. Moreover, compared with *PLD1*^+/+^ mice, *PLD1* depletion in mice enhanced the population of CD4^+^CD25^+^Foxp3^+^Treg cells (51.8% versus 81.1%) and significantly reduced the population of CD25^+^IL-17^+^ Th17 cells (87.0% versus 22.1%) (Appendix A). These results suggested that genetic and pharmacological inhibition of PLD1 promoted the differentiation of Treg cells and suppressed the differentiation of Th17 cells. In addition, we investigated whether the PLD1 inhibitor could regulate the production of IL-17 in the splenocytes. The splenocytes were stimulated with heat-denatured CII, in the presence or absence of the PLD1 inhibitor, and the amount of IL-17 was measured in the supernatant. The PLD1 inhibitor markedly reduced the production of IL-17 in the splenocytes (Appendix A). To further investigate whether PLD1 inhibitor-induced Treg cells could suppress the proliferation of effector cells, we performed a CFSE-based suppression assay. Freshly isolated, CFSE-labeled naïve CD4^+^ T cells were cocultured with induced Treg (iTreg) cells, generated in the presence or absence of the PLD1 inhibitor, at different cell ratios. The proliferation of effector T cells was analyzed by flow cytometry for CFSE dilution. PLD1 inhibitor-induced expanded Treg cells inhibited the proliferation of effector T cells in a cell density-dependent manner (Figure 6E). Additionally, we found that *PLD1* depletion-induced Treg cells were also functional in suppressing the proliferation of effector cells, as analyzed by an in vitro Treg cell suppression assay (Appendix A). These results suggested that the cells induced by the inhibition and depletion of PLD1 were bona fide Treg cells. During differentiation of Th17 from mouse naïve CD4^+^ T cells, the PLD1 inhibitor significantly decreased the secretion and expression of IL-17, as analyzed by ELISA and q-PCR (Figure 6F,G). However, the PLD1 inhibitor markedly increased the expression of Foxp3 under a Treg differentiation condition (Appendix A). These results indicated that pharmacological inhibition of PLD1 increased the population of functional Foxp3^+^ Treg cells and suppressed the population of Th17.

### 2.7. Targeting PLD1 Reduces Osteoclastogenesis and Bone Resorption

Considering that osteoclasts are key cells mediating cartilage damage and bone erosion in RA [25,26], we next examined whether the PLD1 inhibitor could modulate osteoclast activity in CIA mice and in vitro osteoclastogenesis. The sections of joint were stained with tartrate-resistant acid phosphatase (TRAP). Only TRAP-positive multinucleated cells located at the bone surface within the bone erosion areas were considered to be osteoclasts. Multiple TRAP-positive cells were observed on the eroded bone surface in the vehicle-treated CIA mice. The number of osteoclasts in the knee joints of PLD1 inhibitor-treated CIA mice decreased significantly compared with that in vehicle-treated CIA mice (Figure 7A). We further investigated whether the PLD1 inhibitor could affect in vitro osteoclastogenesis. The PLD1 inhibitor suppressed RANKL/M-CSF-induced osteoclast formation from bone marrow macrophages (BMM), as visualized by TRAP staining (Figure 7B). Moreover, the PLD1 inhibitor significantly decreased the number of mature osteoclasts and TRAP activity (Figure 7C,D). The main function of osteoclasts is bone resorption; therefore, we assessed the effect of PLD1 on bone resorption (Figure 7E). The relative bone resorption area and pit number were significantly reduced in PLD1 inhibitor-treated BMMs cultured on bone cell culture system plates (Figure 7F,G). Furthermore, we investigated whether *PLD1* depletion could affect osteoclastogenesis and bone resorption. *PLD1* depletion significantly inhibited osteoclast differentiation from BMMs, with reduced mature osteoclasts and TRAP activity (Figure 7H,I). In addition, *PLD1* ablation also decreased bone resorption with decreased bone resorption area and pit number (Figure 7J,K). These data suggest that pharmacological and genetic targeting of PLD1 suppressed osteoclastogenesis and bone resorption.

## 3. Discussion

Here, we have shown that PLD1 is required for CIA, as genetic and pharmacological inhibition of PLD1 causes suppression of CIA symptoms, such as induction of the inflammatory response, bone destruction, and osteoclastogenesis. The treatment of RA has focused on the remission of the immune reaction in an attempt to halt bone erosion through the inhibition of immune inflammation [27,28,29]. Thus, drugs that can modulate both inflammatory reactions and bone structure might be suitable candidates for RA treatment. PLD1 is considered to be a promising therapeutic target for the treatment of various autoimmune and degenerative diseases [6,7,8,12,13,14,15]. Although PLD1 has been reported to be upregulated in IL-15-induced RASF, IL-1β-stimulated synoviocytes, and synovia from patients with RA [14,30,31], the functional role of PLD1 in systemic bone destruction in RA remains poorly defined. Here, we demonstrate that targeting PLD1 effectively ameliorates arthritis in CIA mice, and thus, PLD1 may contribute to joint inflammation in RA. The mechanism regulating elevated production of proinflammatory cytokines in RA is unknown. In RASF, PLD1 inhibition abolished IL-1β-induced expression of proinflammatory mediators by suppressing the binding of NF-κB to the promoter of its target genes [14]. Thus, targeting PLD1 in mice with CIA might suppress the systemic increase in the levels of inflammatory cytokines via inhibition of NF-κB activity. Moreover, the anti-inflammatory effects of PLD1 inhibition appear to occur through its reciprocal regulation of the population of Th17 cells (inflammatory cells) and Treg cells (immunosuppressive cells), because PLD1 inhibition increases the population of Treg cells and suppresses the population of Th17 cells. Recently, PLD4 was shown to be associated with RA, systemic lupus erythematosus, and systemic sclerosis [32,33,34], suggesting a role of PLD4 in autoimmune diseases. However, PLD4 has no phospholipase activity but is a 5′ exonuclease [35]. Anti-CII antibodies have been recognized as important pathogenic factors in the initiation and development of CIA. The onset and pathogenesis of CIA are associated with a predominance of pathogenic IgG2a because it binds to cartilage and can fix complement [22]. Reduction in IgG2a levels by targeting PLD1 might ameliorate the severity of arthritis, suggesting a potential role of PLD1 in B cells.

Moreover, Treg cells potently suppress osteoclastogenesis and bone resorption [36]. Deficiency of Treg cells exacerbates various experimental autoimmune diseases, including CIA [37]. Thus, Treg cells appear to be an attractive target with substantial therapeutic potential in RA. In fact, Treg adoptive transfer therapy has been used in more than 50 clinical trials on humans. There are highly efficient techniques for the in vitro expansion of autologous Treg cells. The role of PLD1 in Treg differentiation and T-cell activation has been reported by several groups [17,18,19,20]. Inhibition of PLD signaling by 1-butanol leads to enrichment of Foxp3^+^ Treg cells [17] and suppresses the surface expression of CTLA-4, an essential protein in the regulation of T-cell response [18]. Recent studies using small molecule of PLD1-specific inhibitor also show that PLD1 is required for T-cell receptor-mediated signaling, T-cell activation, and effector function during immune response [19,20]. RA and CIA are T-cell-driven diseases and the protection against the diseases is related to the inability of T effectors to be activated. Development of Treg cells is an essential requirement for effective clinical intervention. Therefore, it is speculated that the effect of PLD1 targeting could be due to promotion of Treg differentiation and systemic disruption of T effector cells (Th1 and Th17 cells), therefore rendering attenuating the onset of CIA. In RA patients and the CIA mouse model, a decrease in Treg cells is associated with increased osteoclast formation and bone resorption, and Treg cells can suppress osteoclast differentiation and bone resorption, and prevent the development of CIA [36,38,39]. Only a few studies have shown the effects of PLD in bone cells. PLD is involved in the induction of the preresorptive cytokine, IL-6, in osteoblasts [40] and is required for lung cancer-derived IL-8-induced osteoclastogenesis [41] and IL-15-mediated osteoclastogenesis in RASF [30]. 

These studies suggest that PLD can mediate the inflammatory response and induce bone resorption by stimulating osteoclast differentiation. Identification of the PLD signaling pathway as the crucial regulator of osteoclastogenesis will lead to a better understanding of the mechanisms regulating osteoclastogenesis. At present, it is unknown whether PLD1 regulates osteoclastogenesis and bone resorption via modulation of the Treg/Th17 balance. However, it is possible that regulation of Treg/Th17 cells by targeting of PLD1 in CIA mice might be involved in the suppression of osteoclastogenesis and bone loss. Further studies are needed to investigate the role of PLD1-mediated Treg/Th17 balance in osteoclastogenesis and bone resorption. In conclusion, our data suggest that PLD1 may represent a novel therapeutic target for suppressing immune response, inflammation, and bone loss in autoimmune diseases. The findings of the current study have important implications regarding the potential application of the PLD1 inhibitor for the treatment of RA.

## 4. Materials and Methods

### 4.1. Reagents

PLD1-selective inhibitor, VU0155069, was purchased from Cayman Chemical Co. (Ann Arbor, MI, USA). Dimethylsulfoxide (DMSO) and hematoxylin-erosion solution was purchased from Sigma-Aldrich (St. Louis, MO, USA).

### 4.2. Mice and CIA

Male WT and *PLD1*^−/−^ DBA/1J mice were used to generate the CIA mouse models. *PLD1*^−/−^ mice with a C57BL/6-background were backcrossed with DBA/1J mice to generate the *PLD1*^−/−^ DBA/1J mice (*n* > 7 generations, *n* = 15 per group). All mice were used in accordance with protocols approved by the Animal Care and Ethics Committees of Pusan National University. Mice were injected intradermally at the base of the tail with 100 μg bovine type II collagen (CII) (Chondrex, Redmond, WA, USA) containing complete Freund’s adjuvant (Chondrex). Three weeks later, the mice were given boosters in the tail with 100 μg of CII, emulsified in incomplete Freund’s adjuvant (Chondrex). For efficacy, the PLD1 inhibitor (5 mg/kg) was intraperitoneally injected every other day after arthritis onset on day 21. The incidence and severity of arthritis were evaluated on the indicated days after the first immunization. Severity was evaluated using a clinical score (grade 0–4) of paw swelling. The scale of the arthritis index ranged from 0–4, with a maximum score of 4 for each paw, as described previously [22]. The clinical severity of arthritis was graded as follows: 0 = no evidence of erythema or swelling, 1 = erythema and mild swelling confined to the midfoot or ankle joint, 2 = slight edema or erythema in at least some digits, 3 = moderate edema involving the entire paw, and 4 = severe edema and erythema involving the entire paw and subsequent ankylosis. The clinical score for each mouse was the sum of the scores in each of the 4 paws, with a maximum score of 16. Two independent observers, without prior knowledge of the experimental groups, performed the scoring. Joint tissues were fixed, decalcified in 0.5 M EDTA, embedded in paraffin, and sectioned at 5 μm thickness. Synovitis was evaluated by H&E staining and histopathological changes were scored by independent observers blinded to animal genotype as follows: synovial inflammation (0–4), bone erosion (0–4), cartilage damage (0–4), and leukocyte infiltration (0–4). All animal experiments were approved by the Institutional Animal Care Committee of Pusan National University (PNU-2009-0067, approved on 11 January 2010).

### 4.3. Micro-CT

The knee joints and hind paws of mice (*n* = 15 per group) at day 42 or 45 after immunization of each group were examined using an in vivo micro-CT system (NFR Polaris-G90, Nano Focus Ray, Korea). Briefly, knee joints and hind paws of the mice were scanned and examined by three-dimensional surface rendering with a common threshold and optimized using histomorphometric techniques (NFR Polaris-G90 MV Control, Nano Focus Ray, Korea). Each femur was scanned parallel to its lateral axis using the Sky Scan-1172 (SkyScan, Aartselaar, Belgium). A core of 200 slides, each 11 μm thick (7 mm long), was used for bone morphometry evaluations with the CTAn (CT analyzer) software (SkyScan).

### 4.4. Measurement of Type II Collagen-Specific Autoantibodies

The amount of murine CII-specific autoantibodies in the serum was measured using the Mouse Anti-Type II Collagen IgG Subtype Antibody Assay Kit (Chondrex), according to the manufacturer’s instructions. In brief, microtiter plates were coated with 0.5 μg/well murine CII and incubated with serially diluted test sera. Bound IgG was detected by incubation with horseradish peroxidase (HRP)-conjugated anti-mouse IgG1 or IgG2a and tetramethylbenzidine (TMB) substrate. Absorbance (450 nm) was measured with an ELISA plate reader.

### 4.5. Cytokine Analysis

Serum and secreted levels of TNF-α, IL-1β, IL-6, IFN-γ, MCP-1, and IL-17 were determined using the MILLIPLEX kit (Millipore, Billerica, MA, USA), according to the manufacturer’s instructions.

### 4.6. q-PCR

Total mRNA was isolated. Reverse transcription was done with reverse transcriptase according to the manufacturer’s protocol. IL-17 and Foxp3 expression was then amplified on a thermal cycler (C1000; Bio-Rad, Hercules, CA, USA) with the SYBR Green Supermix (Bio-Rad) as the interaction agent. Quantification analysis of mRNA was normalized with mGAPDH as the housekeeping gene.

### 4.7. Flow Cytometric Analysis

Forty-two days after the animals received primary immunization, their spleens (*n* = 6 per group) were dissected and washed twice with PBS for cell preparation. The spleens were minced, and the red blood cells were lysed with 0.83% ammonium chloride. The cells were filtered through a cell strainer and centrifuged at 2000 rpm at 4 °C for 5 min. The cell pellets were resuspended in RPMI 1640 medium and plated on 96-well plates (BD Biosciences, San Diego, CA, USA) at a concentration of 2 × 10^5^ cells/well. For CD4 and CD25 analysis, splenocytes were stained with FITC-conjugated CD4 (Thermo Fisher Scientific), incubated in Cytofix/Cytoperm (BD Biosciences) and stained with APC-labeled anti-CD25 (Thermo Fisher Scientific) for the detection of T regulatory cells (Treg), according to the manufacturers’ instructions. For intracellular staining, single-cell suspensions were stimulated for 3 d with 50 μg/mL of T-cell proliferation grade CII (Chondrex). GolgiStop (BD Biosciences) was added for the final 3 h of the culture. Cells were stained extracellularly with anti-CD4, fixed and permeabilized with Perm/Fix solution (Thermo Fisher Scientific), and stained intracellularly with anti-IFN-γ, anti-IL-17, and anti-Foxp3 (all from Thermo Fisher Scientific). Directly conjugated isotype-matched rat anti-mouse antibodies (Thermo Fisher Scientific) were used as controls for non-specific staining. Analysis by flow cytometry was performed using CYTOMICS FC500 and CXP Analysis software (Beckman Coulter, Fullerton, CA, USA).

### 4.8. In Vitro Differentiation of Th Cells

CD4^+^ T cells were isolated from the spleen of *PLD1*^+/+^ or *PLD1*^−/−^ mice using the CD4^+^ T cell Isolation Kit (Miltenyi Biotec, Bergisch-Gladhach, Germany). For differentiation of Treg cells, naïve CD4^+^ T cells were plated on anti-mouse CD3 (5 μg/mL; Biolegend, San Jose, CA, USA), anti-mouse CD28 (1 μg/mL; Biolegend), mouse-TGF-β (5 ng/mL; R&D systems, Minneapolis, MN, USA), IL-2 (20 ng/mL; R&D systems), anti-mouse IL-4 (10 ng/mL; Biolegend), and anti-mouse IFN-γ (10 ng/mL; Biolegend) and incubated for 5 d. For differentiation of Th17 cells, CD4^+^ T cells were seeded with mouse IL-6 in Treg differentiation media. The naïve CD4^+^ T cells were treated with 10 μM of PLD1 inhibitor during differentiation of Th17 cells. On day 5, cells were stimulated with 50 ng/mL phorbol 12-myristate 13-acetate (Sigma-Aldrich) and 1 μg/mL ionomycin (Sigma-Aldrich), in the presence of monensin (Thermo Fisher Scientific) for 4 h to determine cytokine expression by flow cytometry on day 5. For dead cell exclusion, the cells were resuspended in PBS containing 5 μL of 7-ADD (BD Biosciences)/1× 10^6^ cells for 10 min, and washed twice. Following surface stain, the cells were incubated with anti-mouse CD16/CD32 (Fc blocker, Thermo Fisher Scientific), anti-mouse CD4 (Thermo Fisher Scientific), and anti-mouse CD25-APC antibodies (Thermo Fisher Scientific) for 20 min at 4 °C. Intracellular staining was performed using a fixation/permeabilization kit (BD Biosciences), anti-mouse Foxp3-PE (Thermo Fisher Scientific), and anti-mouse IL-17-PE (Thermo Fisher Scientific), according to the manufacturer’s instructions.

### 4.9. In Vitro Treg Cell Suppression Assay

Naïve CD4^+^ T cells were differentiated into Treg cells for 5 d in the absence or presence of the PLD1 inhibitor (10 μM). Naïve CD4^+^ T cells were isolated from the spleen of *PLD1*^+/+^ or *PLD1*^−/−^ mice. Differentiated Treg (iTreg) cells were suspended in 2 × 10^6^ cells/mL with PBS containing 1% FBS, and treated with 50 μg/mL of mitomycin C (Sigma-Aldrich) for 30 min, washed three times with PBS, resuspended with 20 mL of RPMI1640 containing 10% FBS, incubated for 2 h, washed two times with the media, and resuspended at 5 × 10^5^ cells/mL of RPMI1640 containing 10% FBS. The iTreg cells were mixed thoroughly and 50 μL was titrated into the next well to generate serial dilutions in 1 μg/mL of anti-mouse CD3ε and anti-CD28 coated 96-well round bottom plate. Isolated naïve CD4^+^ T cells were suspended at 2 × 10^6^ cells/mL, treated with 4 μM of carboxyfluorescein succinimidyl ester (CFSE; Sigma-Aldrich), and incubated in the dark without agitation for 10 min. CFSE-labeled naïve CD4^+^ T cells were washed with PBS containing 1% FBS and resuspended at 5 × 10^5^ cells/mL in RPMI1640 containing 10% FBS. CFSE-labeled naïve CD4^+^ T cells (100 μL) were added to suppression assay plates and incubated for 3 d. Proliferation was analyzed by staining with anti-mouse CD4-PE (Thermo Fisher Scientific) and CFSE dilution using flow cytometry.

### 4.10. Immunofluorescence Staining

Synovia from mice knees were fixed in 4% paraformaldehyde (PFD) for 48 h, decalcified in 10% EDTA for 1 week, and then kept in 30% sucrose until they were cut into 7 μm thick sections. Antigen retrieval was performed by microwave treatment for 10 min in 10 mM citrate buffer solution (pH 6.0). Sections were deparaffinized, rehydrated, and treated with 10% goat blocking serum for 20 min and incubated with anti-TNF-α (Abcam, Cambridge, MA, USA) overnight at 4 °C. After rinsing with PBS for 15 min, tissues were incubated with goat anti-rabbit Alexa Fluor 568 (BD Biosciences) at room temperature. Spleens from mice were collected, embedded in Tissue-Tek Optimal Cutting Temperature compound (Sakura Finetek, Nagano, Japan), and snap-frozen in liquid nitrogen. Cryosections (7 μm thick) were fixed with 4% PFD, blocked with 10% goat serum for 20 min, and stained using FITC-conjugated anti-Foxp3 and APC-conjugated anti-CD25 (all from eBioscience) overnight at 4 °C. Nuclei were counterstained with 2 μM 4′, 6-diamidino-2-phenylindole (DAPI; Thermo Fisher Scientific) for 5 min. Thereafter, sections were washed twice in PBS for 10 min and twice in distilled water for 10 min, mounted in ProLong Gold anti-fade reagent (Cell Signaling Technology, Boston, MA, USA), and imaged using an inverted fluorescence microscope (EVOS FL Cell Imaging System; Life Technologies, Darmstadt, Germany).

### 4.11. TRAP Staining

Tissues or cells were fixed and stained with TRAP (Sigma-Aldrich) for 1 h at 37 °C, followed by counterstaining with a hematoxylin solution. TRAP^+^ multinuclear cells (MNCs) (with more than three nuclei) were regarded as osteoclasts and counted under an inverted phase contrast microscope. Morphological features of osteoclasts were photographed with a photomicroscope. For detecting the osteoclasts in the tissue sections, osteoclast numbers were measured by quantifying cells positively stained for TRAP. Briefly, specimens were fixed for 30 s and then stained with naphthol AS-BI phosphate and tartrate solution for 1 h at 37 °C, followed by counterstaining with hematoxylin. TRAP-positive multinuclear cells with three or more nuclei were regarded as osteoclasts and counted under an inverted phase contrast microscope. To analyze osteoclasts in joint tissues, each joint section was treated sequentially with a commercial acid phosphatase kit (Sigma-Aldrich), which was used to detect the TRAP enzyme. The osteoclasts in an entire field (magnification 200×) were counted.

### 4.12. In Vitro Osteoclastogenesis Assay

For differentiation of osteoclasts, bone marrow cells from mouse were isolated and cultured with 10% FBS DMEM containing 30 ng/mL of mouse M-CSF (R&D systems) and 50 ng/mL mouse RANKL (R&D systems), in the absence or presence of the PLD1 inhibitor for 6 d. TRAP staining was performed with the aid of a commercial kit (Sigma-Aldrich), according to the manufacturer’s instructions. TRAP-positive multinucleated cells containing three or more nuclei were counted as mature osteoclasts. TRAP activity was determined as previously described [42]. For the pit formation assay, osteoclasts were differentiated as mentioned above, in BioCoat OSTEOLOGIC^TM^ Bone Cell Culture System plates (BD Biosciences) that had been coated with a thin film of calcium phosphate for 5 d. The cells were removed, and total resorption pits were observed under a bright-field microscope as clear zones in the matrix. The resorbed areas on the plates were captured with a digital camera attached to the microscope and analyzed by a soft imaging system. The percentage of resorption was calculated by considering the area of a single well as 100% with six to eight wells per group. These experiments were repeated independently at least three times.

### 4.13. Statistical Analysis

The data are presented as the mean ± standard error of the mean from three or more independent experiments. Statistical analyses were performed by one-way analysis of variance or Student’s *t*-test. *p* values of < 0.05 were considered significant.

## Figures and Tables

**Figure 1 ijms-21-03230-f001:**
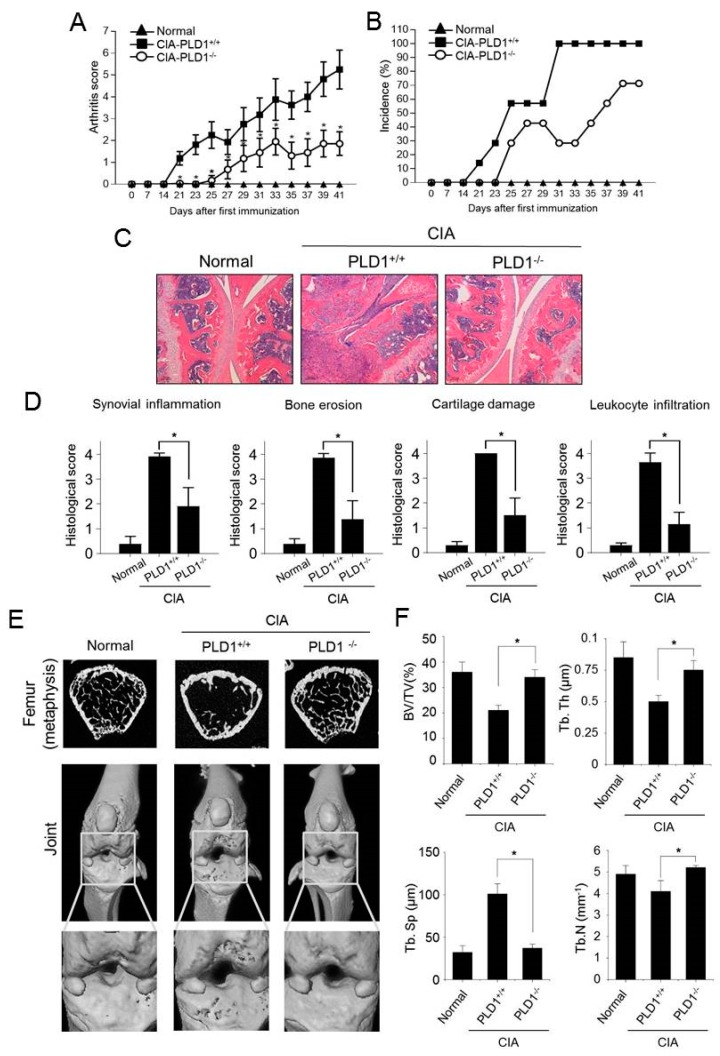
*PLD1* deficiency reduces disease severity and joint inflammation in collagen-induced arthritis (CIA) mice. (**A**) Clinical arthritis scores were determined in CIA-challenged *PLD1*^+/+^ and *PLD1*^−/−^ mice. (**B**) Incidence of CIA, as scored in all paws. (**C**) Hematoxylin and eosin staining of knee joint sections from the indicated mice at day 42. (**D**) Histological assessment of the pathological features of mice knee joints at day 42. (**E**) Representative micro-computerized tomography images of femurs and knee joints of mice at day 42 (top—transverse sections of metaphysis; middle—three-dimensional reconstructed knee joint; bottom—magnified view) (**F**) Histomorphometric analysis of the bone for each group, showing bone volume (BV)/tissue volume (TV) ratio, trabecular bone thickness (Tb.Th), trabecular separation (Tb.Sp), and trabecular number (Tb.N). *n* = 15 per group. * *p* < 0.05. Quantitative results are shown as mean ± standard error of the mean (SEM).

**Figure 2 ijms-21-03230-f002:**
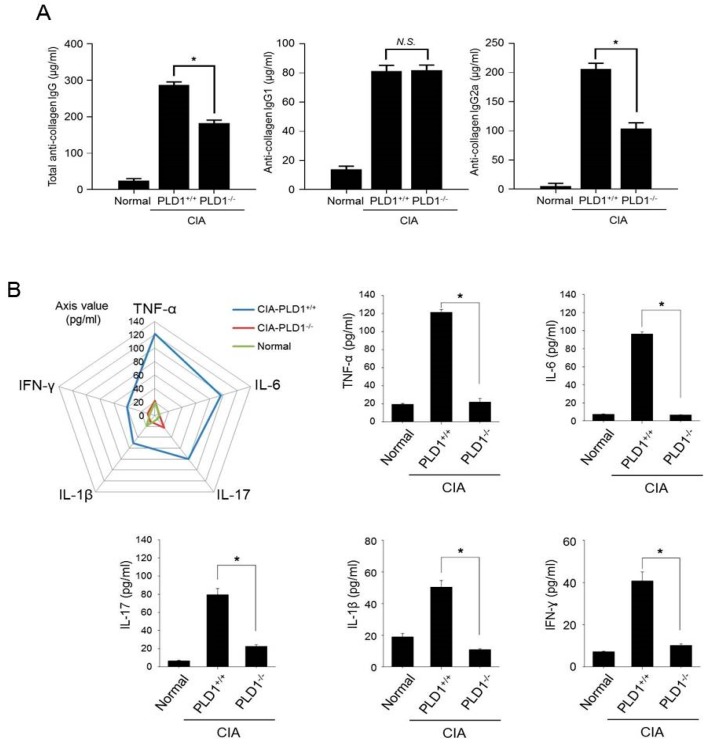
*PLD1* deficiency suppresses the collagen type II-specific humoral response and the production of proinflammatory cytokines in collagen-induced arthritis (CIA) mice. (**A**) The amount of anti-collagen total IgG, IgG1, and IgG2a antibodies was measured by ELISA in the serum from the indicated mice at day 42. (**B**) Measurement of proinflammatory cytokines in the serum of the indicated mice at day 42 as analyzed by ELISA. *n* = 15 per group. * *p* < 0.05, N.S., non-significant. Results are shown as mean ± standard error of the mean (SEM).

**Figure 3 ijms-21-03230-f003:**
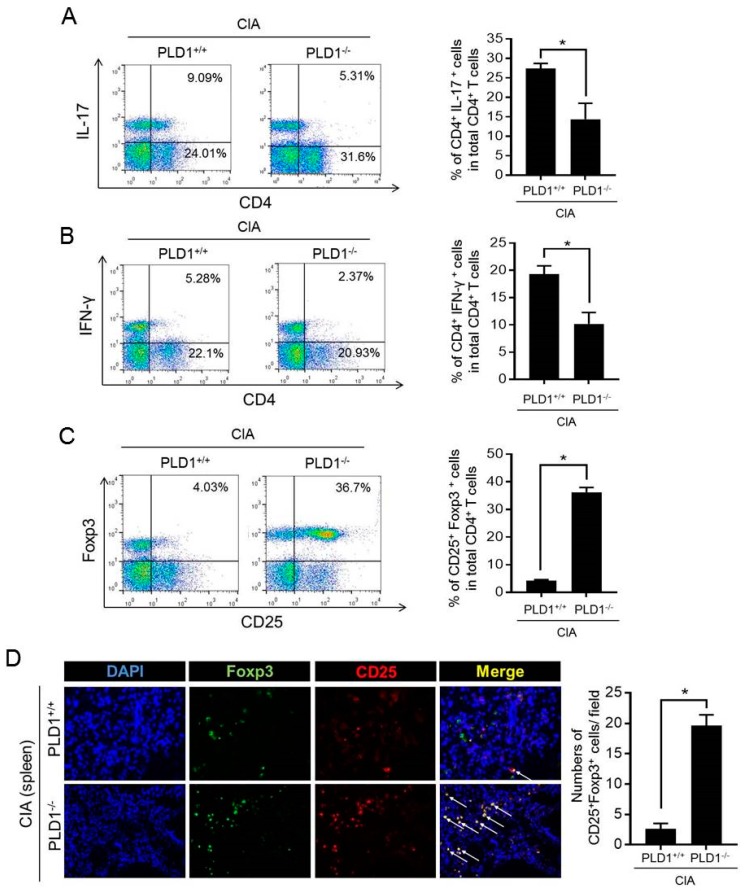
*PLD1* deficiency decreases the population of Th1/Th17 cells and increases the population of Treg cells in collagen-induced arthritis (CIA) mice. (**A**) Representative flow cytometry profile of IL-17^+^ cell frequency among CD4^+^ T cells in the spleens of indicated mice with CIA (left); quantitative analysis of the CD4^+^IL-17^+^ cell population in spleen CD4^+^ T cells from mice with CIA (right). (**B**) Representative flow cytometry profile of CD4^+^IFN-γ^+^ T-cell frequency in the spleens of indicated mice (left); quantitative analysis of the CD4^+^IFN-γ^+^ T-cell population (right). (**C**) Representative flow cytometry profile of CD25^+^Foxp3^+^ cell frequency among splenic CD4^+^ T cells (left); quantitative analysis of the CD25^+^Foxp3^+^ Treg cell population in splenic CD4^+^ T cells (right). (**D**) Immunofluorescence staining of Foxp3 and CD25 in the spleen of the indicated mice with CIA (left); quantification CD25^+^Foxp3^+^ cell number. Treg cells are indicated by arrows. Original magnification, × 400. 3 fields per mouse spleen, 6 mice per group. * *p* < 0.05. The analysis was performed in CIA-challenged *PLD1*^+/+^ and *PLD1*^−/−^ mice at day 42. Results are representative of at least six independent experiments and are shown as mean ± standard error of the mean (SEM).

**Figure 4 ijms-21-03230-f004:**
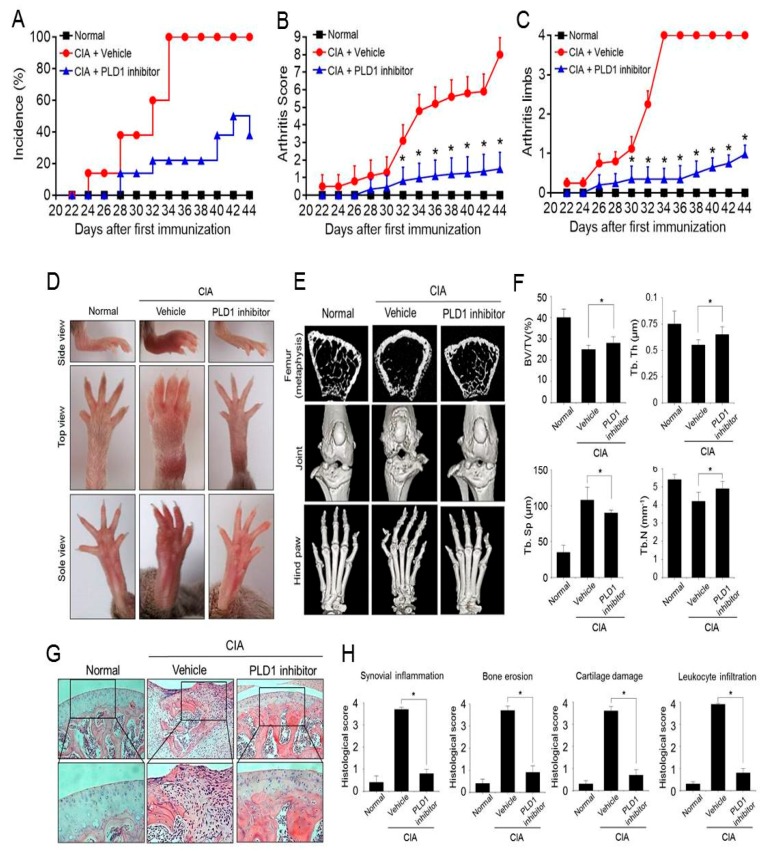
The PLD1 inhibitor ameliorates arthritis and bone erosion in collagen-induced arthritis (CIA) mice. (**A**–**C**) Effect of the PLD1 inhibitor on the incidence, arthritic score, and the number of affected limbs of CIA mice. DBA/1J mice were immunized on day 0 and boosted on day 21 with type II collagen; intraperitoneal injection of the vehicle or PLD1 inhibitor (5 mg/kg) was done every other day start on day 21 post immunization. The clinical symptoms of the disease including incidence (**A**), arthritic score (**B**), and the number of affected limbs (**C**) were scored. (**D**) Representative photographs showing the gross features of the hind paws at day 45. (**E**) Representative micro-computerized tomography (CT) images of the femurs and knee joints of mice at day 45. (**F**) Effect of the PLD1 inhibitor on the micro-CT parameters of trabecular bone mass in CIA mice at day 45. (**G**) Hematoxylin and eosin staining of knee joints from the indicated mice at day 45. Boxed areas are shown at approximately two times higher magnification. (**H**) Histological assessment of knee joints. *n* = 15 per group. * *p* < 0.05. Quantitative results are shown as mean ± standard error of the mean (SEM).

**Figure 5 ijms-21-03230-f005:**
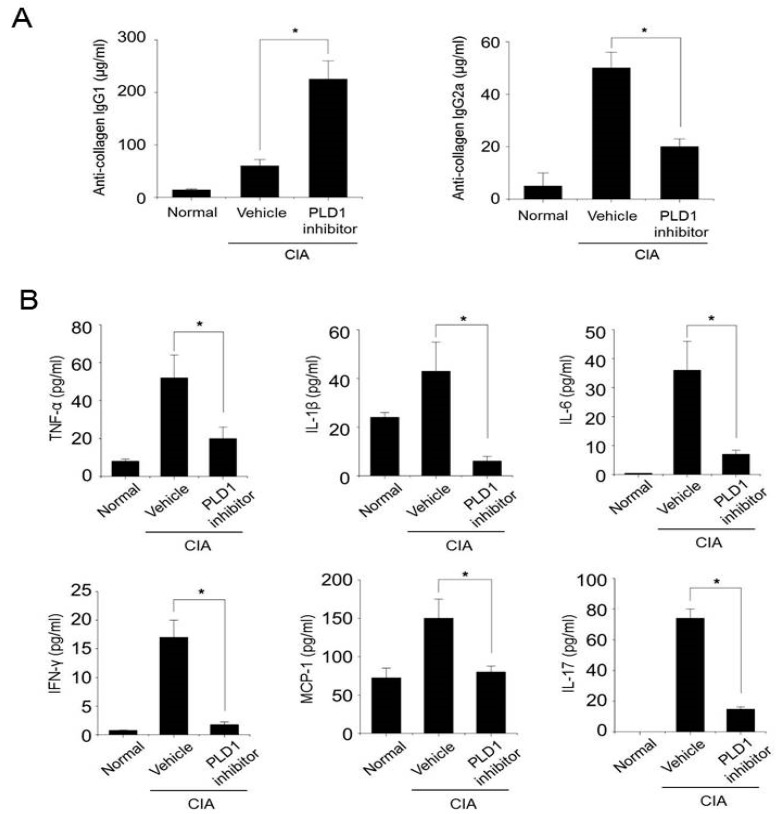
The PLD1 inhibitor reduces the production of the anti-CII IgG2a autoantibody and proinflammatory cytokines in collagen-induced arthritis (CIA) mice. (**A**) The amount of anti-collagen IgG1 and IgG2a antibodies was measured by ELISA in the serum of the indicated mice at day 45. (**B**) The amount of proinflammatory cytokines was measured by ELISA in the serum of the indicated mice at day 45. *n* = 15 per group. * *p* < 0.05. Results are shown as mean ± standard error of the mean (SEM).

**Figure 6 ijms-21-03230-f006:**
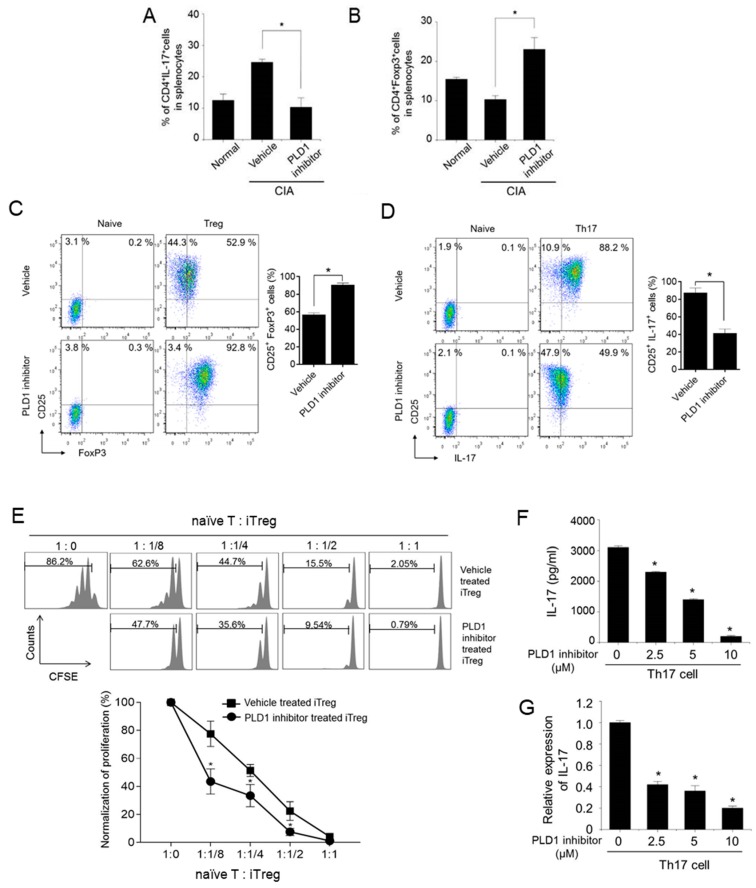
Pharmacological inhibition of PLD1 modulates the Th17/Treg balance in murine splenocytes. (**A**) Effect of the PLD1 inhibitor on the population of CD4^+^IL-17^+^ cells in the spleens of the indicated mice. (**B**) Effect of the PLD1 inhibitor on the population of CD4^+^Foxp3^+^ cells in the spleen of mice with CIA. (**C**,**D**) Effect of the PLD1 inhibitor on in vitro differentiation of Treg (**C**) and Th17 (**D**) cells. Naïve CD4^+^ T cells isolated from mice spleens were cultured under condition of Treg cells or Th17 cell differentiation. Representative flow cytometry profile and quantification for CD25^+^Foxp3^+^ Treg and CD25^+^IL-17^+^ Th17 cell populations. (**E**) Freshly isolated naïve CD4^+^ T cells (naïve T cells), labeled with CFSE were cocultured in the presence of anti-mouse CD3ε and anti-mouse CD28 for 72 h with induced Treg cells (iTreg) generated in the presence or absence of the PLD1 inhibitor. Proliferation of naïve CD4^+^ T cells was analyzed by flow cytometry for CFSE dilution. (**F**,**G**) Effect of the PLD1 inhibitor on the secretion (**F**) and expression (**G**) of IL-17 during differentiation of Th17 cells as analyzed by ELISA and q-PCR, respectively. *n* = 6 per group. * *p* < 0.05. Results are representative of at least six independent experiments, and shown as mean ± standard error of the mean (SEM).

**Figure 7 ijms-21-03230-f007:**
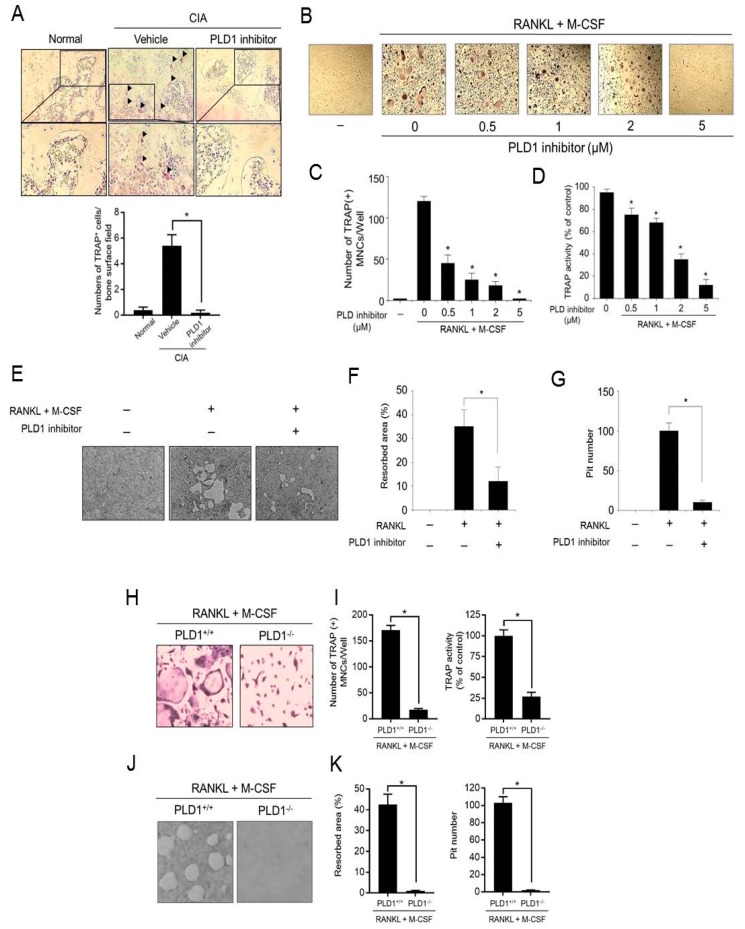
Targeting of PLD1 reduces osteoclastogenesis and bone resorption. (**A**) Representative histological images of tartrate-resistant acid phosphatase (TRAP) staining for osteoclasts (red) at a region distant to the site of inflammation in the proximal tibia of mice from different groups at day 45. (**B**) Representative TRAP staining images of osteoclasts differentiated by RANKL and M-CSF induction in the presence of the indicated concentration of the PLD1 inhibitor in murine bone marrow cells (BMMs). (**C**,**D**) The cells were stained with TRAP and the number of multinucleated (nuclei ≥ 3) TRAP^+^ cells (**C**) and TRAP activity (**D**) were counted. (**E**) Representative images of bone resorption assay in differentiated osteoclasts. (**F**,**G**) Effect of the PLD1 inhibitor on the relative bone resorption area (**F**) and pit number (**G**). Representative TRAP staining images (**H**) and TRAP-positive cell numbers and TRAP activity (**I**) of the osteoclasts differentiated from BMMs from *PLD1*^+/+^ and *PLD1*^−/−^ mice. Representative bone resorption images (**J**) and resorbed area and pit number (**K**) of the osteoclasts differentiated from BMMs from *PLD1*^+/+^ and *PLD1*^−/−^ mice. *n* = 6 per group. * *p* < 0.05. Results are representative of at least six independent experiments and are shown as mean ± standard error of the mean (SEM).

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
