# Peer review of "Targeting of Phospholipase D1 Ameliorates Collagen-Induced Arthritis via Modulation of Treg and Th17 Cell Imbalance and Suppression of Osteoclastogenesis"

_ijms, 2020, doi:10.3390/ijms21093230_

Round 1
Reviewer 1 Report
In this paper, the authors show an impact of PLD1 in the prevention of CIA. They show that inhibition of PLD1 genetically and chemically reduce the severity of CIA. The authors investigated the cellular event underlying the clinical outcome and showed that PLD1 inhibition leads to downregulation of cytokine and antibodies production and shift the balance Th17/Tregs toward tolerance. Although the data are clear and statistically significant, there are some missing points in the interpretations/discussion of the results. Firstly, the role of PLD1 in Treg differentiation and upregulation of FOXP3 has been already reported (DOI:https://doi.org/10.1038/nmeth903; DOI:10.4049/jimmunol.174.8.4803). Secondly, this report disregards the major role of PLD1 in T cell activation signaling and biology that has been broadly studied by several groups. (10.1046/j.1365-2567.1997.00150.x; PMID:8077660; DOI:10.1371/journal.ppat.1004864; DOI:10.4049/jimmunol.1701291). Here, the effect of PLD1 inhibition could be due to the systemic disruption of T cell activation in the mouse therefore rendering impossible or attenuating the onset of CIA. The data should be discussed in regards to the previously published works.
Minor comments:
- Line 32: CIA instead of “CIA induced arthritis”
- Line 41: typo for the reference?
- Line 354: (typo) reagents instead of regent
- L342: The authors state that “However, therapeutic applications of Treg cells are limited because of their scarcity” but Treg adoptive transfer therapy are used in human in more than 50 clinical trials. There are very efficient techniques to in vitro expand autologous Tregs. Therefore, this above statement could be updated/re-phrased.
- Fig 3: The authors could look at other Treg markers (CTLA4, GARP).
- Fig 6: C and D: The intracellular staining for FOXP3 and IL17 are not convincing. the whole population has shifted. It would be good if the authors could provide FSC and SSC as well as the gating strategy used for those 2 stainings.
Major comments:
- RA and CIA are T cell driven diseases but the authors disregard the function of PLD1 in T cell activation. The protection against CIA is related to an inability of T effectors to be activated.
- Effect observed in B cells could also be due to an inefficient inflammatory response from the T cell subset.
- Fig6E: The protocol used for the Treg suppression assay is not clear. If PLD1 inhibitor remains in the culture, it could inhibit T effector proliferation. Could this be done with PLD1 KO Tregs?
Author Response
Response to Reviewer 1 Comments
Point 1: In this paper, the authors show an impact of PLD1 in the prevention of CIA. They show that inhibition of PLD1 genetically and chemically reduce the severity of CIA. The authors investigated the cellular event underlying the clinical outcome and showed that PLD1 inhibition leads to downregulation of cytokine and antibodies production and shift the balance Th17/Tregs toward tolerance. Although the data are clear and statistically significant, there are some missing points in the interpretations/discussion of the results. Firstly, the role of PLD1 in Treg differentiation and upregulation of FOXP3 has been already reported (DOI:https://doi.org/10.1038/nmeth903; DOI:10.4049/jimmunol.174.8.4803). Secondly, this report disregards the major role of PLD1 in T cell activation signaling and biology that has been broadly studied by several groups. (10.1046/j.1365-2567.1997.00150.x; PMID:8077660; DOI:10.1371/journal.ppat.1004864; DOI:10.4049/jimmunol.1701291). Here, the effect of PLD1 inhibition could be due to the systemic disruption of T cell activation in the mouse therefore rendering impossible or attenuating the onset of CIA. The data should be discussed in regards to the previously published works.
Response 1: Thank you very much for your kind comments. As you comment, we described the previously published works in the Introduction and Discussion section. In the Introduction section (Line 62-63), we added the following sentence “PLD1 has been reported to plays an important role in Treg differentiation and T-cell activation [17-20]”. In the Discussion section, we discussed in regards to the previously published works. We added following sentences/paragraphs (Line 348-363): ” The role of PLD1 in Treg differentiation and T-cell activation has been reported by several groups [17-20]. Inhibition of PLD signaling by 1-butanol leads to enrichment of Foxp3+ Treg cells [17] and suppresses the surface expression of CTLA-4, an essential protein in the regulation of T-cell response [18]. Previously, 1-butanol has been widely employed to identify PLD/PA-driven processes. However, 1-butanol does not always effectively reduce PA accumulation, and its use may result in PLD-independent, deleterious effects [39]. Several biological processes suppressed by 1-butanol are not affected by the small-molecule, PLD inhibitor [39]. It has also been suggested that reduced CTLA-4 expression by 1-butanol is a consequence of a nonspecific effect [19]. Recent studies using small molecule of PLD1-specific inhibitor show that PLD1 is required for T-cell receptor-mediated signaling, T-cell activation, and effector function during immune response [19,20]. RA and CIA are T-cell-driven diseases and thus PLD1 might play a pivotal role in T cell activation. The protection against CIA is related to the inability of T effectors to be activated. Therefore, development of Treg cells is an essential requirement for effective clinical intervention. It is speculated that the effect of PLD1 targeting could be due to promotion of Treg differentiation and systemic disruption of T effector cells (Th1 and Th17 cells), therefore rendering attenuating the onset of CIA.”.
Point 2: Line 32: CIA instead of “CIA induced arthritis”
Response 2: We very thank helpful your comments. We rewrote CIA instead of “CIA-induced arthritis. (Line 29 in revised version). We also fixed Th17 instead of T17 in Line 25, 27 and 30 of revised version.
Point 3: Line 41: typo for the reference?
Response 3: We very thank for your kind comment. We fixed it (Line 36 in revised version).
Point 4: Line 354: (typo) reagents instead of regent
Response 4: We very thank for your kind comment. We rewrote reagents instead of regent.
(Line 382 in revised version)
Point 5: The authors state that “However, therapeutic applications of Treg cells are limited because of their scarcity” but Treg adoptive transfer therapy are used in human in more than 50 clinical trials. There are very efficient techniques to in vitro expand autologous Tregs. Therefore, this above statement could be updated/re-phrased.
Response 5: Thank you very much for your helpful comments. As you recommend, we re-phased the sentences. The previous description “However, therapeutic applications of Treg cells are limited because of their scarcity” was removed. In the Discussion section, the following sentences were added in Line 346-348 of revised version: “In fact, Treg adoptive transfer therapy has been used in more than 50 clinical trials on humans. There are highly efficient techniques for the in vitro expansion of autologous Treg cells”.
Point 6: The authors could look at other Treg markers (CTLA4, GARP).
Response 6: Thank you very much for your kind comments. As you recommend, other Treg markers can be observed. In this study, we analyzed Treg cells using Foxp3, a typical marker of Treg cells, although there are other Treg markers such as CTLA4 and GARP. We added this sentence in the result of Fig. 3 (Line 160-161 of revised version).
Point 7: Fig 6: C and D: The intracellular staining for FOXP3 and IL17 are not convincing. the whole population has shifted. It would be good if the authors could provide FSC and SSC as well as the gating strategy used for those 2 stainings.
Response 7: Thank you very much for your kind comments. As you comment, we provided the data of FSC and SSC as well as the gating strategy in Figure S5. We added the following sentences in Result section (Line 255-256 of revised version): “Forward scatter, side scatter, and gating strategy for the staining of intracellular Foxp3 and IL-17 have been shown in Figure S5”.
Point 8: RA and CIA are T cell driven diseases but the authors disregard the function of PLD1 in T cell activation. The protection against CIA is related to an inability of T effectors to be activated. Effect observed in B cells could also be due to an inefficient inflammatory response from the T cell subset.
Response 8: Thank you very much for your helpful comments. As you comment kindly, we added the following paragraphs in Introduction and Discussion section. In the Introduction section (Line 62-63), we added the following sentence: “PLD1 has been reported to plays an important role in Treg differentiation and T-cell activation [17-20]”. In the Discussion section (Line 348-362 of revised version), we added the following paragraph: “The role of PLD1 in Treg differentiation and T-cell activation has been reported by several groups [17-20]. Inhibition of PLD signaling by 1-butanol leads to enrichment of Foxp3+ Treg cells [17] and suppresses the surface expression of CTLA-4, an essential protein in the regulation of T-cell response [18]. Previously, 1-butanol has been widely employed to identify PLD/PA-driven processes. However, 1-butanol does not always effectively reduce PA accumulation, and its use may result in PLD-independent, deleterious effects [39]. Several biological processes suppressed by 1-butanol are not affected by the small-molecule, PLD inhibitor [39]. It has also been suggested that reduced CTLA-4 expression by 1-butanol is a consequence of a nonspecific effect [19]. Recent studies using small molecule of PLD1-specific inhibitor show that PLD1 is required for T-cell receptor-mediated signaling, T-cell activation, and effector function during immune response [19,20]. RA and CIA are T-cell-driven diseases and thus PLD1 might play a pivotal role in T cell activation. The protection against CIA is related to the inability of T effectors to be activated. Therefore, development of Treg cells is an essential requirement for effective clinical intervention. It is speculated that the effect of PLD1 targeting could be due to promotion of Treg differentiation and systemic disruption of T effector cells (Th1 and Th17 cells), therefore rendering attenuating the onset of CIA.”. In Line 342 -343 of revised version, we added the following sentences: “The generation of pathogenic autoantibodies to CII by B cells might also be owing to an inefficient inflammatory response from the T-cell subset”.
Point 9: Fig6E: The protocol used for the Treg suppression assay is not clear. If PLD1 inhibitor remains in the culture, it could inhibit T effector proliferation. Could this be done with PLD1 KO Tregs?
Response 9: As you comment kindly, we described in detail the protocol used for the Treg suppression assay in Material and Methods (Line 469-474 and 481-482 of revised version):
“Naïve CD4+ T cells were differentiated into Treg cells for 5 d in the absence or presence of the PLD1 inhibitor (10 μM). Naïve CD4+ T cells were isolated from the spleen of PLD1+/+ or PLD1−/− mice. Differentiated Treg (iTreg) cells were suspended in 2 × 106 cells/mL with PBS containing 1% FBS, and treated with 50 μg/mL of mitomycin C (Sigma-Aldrich) for 30 min, washed three times with PBS, resuspended with 20 mL of RPMI1640 containing 10% FBS, incubated for 2 h, washed two times with the media, and resuspended at 5 × 105 cells/mL of RPMI1640 containing 10% FBS. The iTreg cells were mixed thoroughly and 50 μL was titrated into the next well to generate serial dilutions in 1 μg/mL of anti-mouse CD3ε and anti-CD28 coated 96-well round bottom plate. Isolated naïve CD4+ T cells were suspended at 2 × 106 cells/mL, treated with 4 μM of carboxyfluorescein succinimidyl ester (CFSE; Sigma-Aldrich), and incubated in the dark without agitation for 10 min. CFSE-labeled naïve CD4+ T cells were washed with PBS containing 1% FBS and resuspended at 5 × 105 cells/mL in RPMI1640 containing 10% FBS. CFSE-labeled naïve CD4+ T cells (100 μL) were added to suppression assay plates and incubated for 3 d. Proliferation was analyzed by staining with anti-mouse CD4-PE (Thermo Fisher Scientific) and CFSE dilution using flow cytometry”.
Since PLD1 inhibitor was removed by washing, the possibility which PLD1 inhibitor inhibits T effector proliferation can be excluded. As you recommend, we have done the Treg suppression assay with PLD1 KO Treg. We added the following sentences in Line 271-274 of revised version: “Additionally, we found that PLD1 depletion-induced Treg cells were also functional in suppressing the proliferation of effector cells, as analyzed by an in vitro Treg cell suppression assay (Figure S8). These results suggest that the cells induced by the inhibition and depletion of PLD1, were bona fide Treg cells”.

Reviewer 2 Report
In the manuscript titled “Targeting of phospholipase D1 ameliorates collagen-induced arthritis via modulation of Treg and Th17 cell imbalance and suppression of osteoclastogenesis” the authors investigate role of PLD1 in CIA using PLD1-knockout mouse and PLD1 inhibitor. The study was well designed and well written. Findings are informative.
The following are some concerns:
- Line 41, citation format is not consistent with rest of the manuscript.
- 1 legend, indicate time point the joint samples for histology and microCT shown in figure 1 were obtained.
- Line 122, “….and complement fixation”. Need reference.
- 2 legend, indicate time point the serum samples for antCII Ab and cytokine ELISA shown in Fig 2 were obtained.
- Lin 147, “CD4+IFNg+ ells” should be CD4+IFNg+
- 3A-3C, in addition to present frequency, present cell # data.
- 3 legend, indicate time point the spleen cells for Th17 vs Treg FACS analysis shown in Fig 3 were obtained.
- Fig 4 legend, add information on PLD1 inhibitor treatment regimen (e.g. when it treated, how many time and what interval, etc.). Also indicate time point the joint samples for histology and microCT shown in figure 4 were obtained.
- 5 legend, indicate time point the serum samples for antCII Ab and cytokine ELISA shown in Fig 5 were obtained.
- Please add the data regarding effects of PLD1 in Th17 differentiation and Treg differentiation and suppressive ability of Treg cells using wile-type vs PLD1KO T cells.
- 7 legend, indicate time point the tibia samples for histology shown in figure 7A were obtained. Also provide information on cell source for in vitro study in Fig 7B-7G.
- Please add the data regarding effects of PLD1 in osteoclastogenesis and bone resorption using wile-type vs PLD1KO BMM cells.
- Based on data in Fig 7, PLD1 plays a critical role in osteoclastogenesis and bone resorption regardless of its Th17/treg balance. Any discussion on regulatory role of PLD1 in osteoclastogenesis and bone resorption?
- Line 365, what is booster material?
- Line 366, provide PLD1 inhibitor treatment regimen.
- Section 4.4. based on what described in this section, it seemed like to this reviewer that the authors measured total IgG1 and IgG2a not CII-specific autoantibodies in this study. Please clarify what the authors really measured in serum. Just IgG1 and IgG2a, or anti-bovine CII-specific IgG1 and IgG2a, or anti-mouse CII-specific IgG1 and IgG2a? Please provide detail procedures of this ELISA. What was coated in the ELISA plate?
- Section 4.7, How many events/samples were analyzed?
- S2, S3, & S5, indicate time of sampling.
- S, S4, indicate sample size.
Author Response
Response to Reviewer 2 Comments
Point 1: Line 41, citation format is not consistent with rest of the manuscript.
Response 1: As you comment kindly, the word “[Lee, 2001 #1]” has been changed to “[1]” in Line 36 of revised version. We also fixed the word “(Bernhardt et al., 2017)” to “[44]” in section 4.12 and added reference [44] in Line 517 of revised version.
Point 2: 1 legend, indicate time point the joint samples for histology and microCT shown in figure 1 were obtained.
Response 2: As you comment kindly, we indicated the time point the joint samples for histology and microCT in the legend of Figure 1 (Line 97-99 of revised version)
Point 3: Line 122, “….and complement fixation”. Need reference.
Response 3: As you comment kindly, As you pointed out kindly, we added the reference [22] in Line 123 of revised version.
Point 4: 2 legend, indicate time point the serum samples for antCII Ab and cytokine ELISA shown in Fig 2 were obtained.
Response 4: As you comment kindly, we indicated the time point the serum samples for anti-CII Ab and cytokine ELISA in the legend of Figure 2. (Line 113-114 of revised version)
Point 5: Lin 147, “CD4+IFNg+ ells” should be CD4+IFNg+
Response 5: As you comment kindly, we changed the word “CD4+IFNg+ ells” to “CD4+IFNg+” and made additional change the word “CD4+IL-17+ ells” to “CD4+IL-17+” in Line 154-155 of revised version.
Point 6: 3A-3C, in addition to present frequency, present cell # data.
Response 6: As you comment kindly, we made change the word “CD4+IL-17+ cells (%)” to “CD4+IL-17+ cells frequency of splenocytes” in figure 3A, “CD4+IFN-ɣ+ cells (%)” to “CD4+IFN-ɣ+ cells frequency of splenocytes” in figure 3B and “CD25+Foxp3+ cells (%)” to “CD25+Foxp3+ cells frequency of CD4+ T cells” in figure 3C.
Point 7: 3 legend, indicate time point the spleen cells for Th17 vs Treg FACS analysis shown in Fig 3 were obtained.
Response 7: As you comment kindly, we indicated the time point the spleen cells for Th17 vs Treg FACS analysis in the legend of Figure 3 (Line 150 of revised version).
Point 8: Fig 4 legend, add information on PLD1 inhibitor treatment regimen (e.g. when it treated, how many time and what interval, etc.). Also indicate time point the joint samples for histology and microCT shown in figure 4 were obtained.
Response 8: As you comment kindly, we added information on PLD1 inhibitor treatment regimen and inserted the time point the joint samples for histology and microCT in the legend of Figure 4. (Line 196-197, 199-202 of revised version)
Point 9: 5 legend, indicate time point the serum samples for antCII Ab and cytokine ELISA shown in Fig 5 were obtained.
Response 9: As you comment kindly, we indicated the time point the serum samples for anti-CII Ab and cytokine ELISA in the legend of Figure 5 (Line 226 and 228 of revised version)
Point 10: Please add the data regarding effects of PLD1 in Th17 differentiation and Treg differentiation and suppressive ability of Treg cells using wile-type vs PLD1KO T cells.
Response 10: As you comment kindly, we added the data in figure S6 for effect of PLD1 in Th17 differentiation and Treg differentiation, and described the following sentences in Line 257-261 of revised version: “Moreover, compared with PLD1+/+ mice, PLD1 depletion in mice enhanced the population of CD4+CD25+Foxp3+Treg cells (29.2% versus 58.7%) and significantly reduced the population of CD25+IL-17+ Th17 cells (40.1% versus 15.2%) (Figure S6). These results suggested that genetic and pharmacological inhibition of PLD1 promoted the differentiation of Treg cells and suppressed the differentiation of Th17 cells”.
We added the data in figure S8 for effect of PLD1 in suppressive ability of Treg cells using PLD1KO T cells, and added the following sentences in Line 271-274 of revised version:” Additionally, we found that PLD1 depletion-induced Treg cells were also functional in suppressing the proliferation of effector cells, as analyzed by an in vitro Treg cell suppression assay (Figure S8). These results suggested that the cells induced by the inhibition and depletion of PLD1 were bona fide Treg cells”.
Point 11: legend, indicate time point the tibia samples for histology shown in figure 7A were obtained. Also provide information on cell source for in vitro study in Fig 7B-7G.
Response 11: As you comment kindly, we indicated the time point the tibia samples for histology and provided information on cell source for in vitro study in Fig 7B-7G. (Line 305, 307-308 of revised version)
Point 12: Please add the data regarding effects of PLD1 in osteoclastogenesis and bone resorption using wile-type vs PLD1KO BMM cells.
Response 12: As you comment kindly, we added the data regarding effects of PLD1 in osteoclastogenesis and bone resorption using wile-type vs PLD1KO BMM cells in Figure 7H-K. Title of Figure 7 was changed into “Targeting of PLD1 reduces osteoclastogenesis and bone resorption”. We added the following pargraphs in Line 296-301 of revised version: “Furthermore, we investigated whether PLD1 depletion could affect osteoclastogenesis and bone resorption. PLD1 depletion significantly inhibited osteoclast differentiation from BMMs, with reduced mature osteoclasts and TRAP activity (Figure 7H,I). In addition, PLD1 ablation also decreased bone resorption with decreased bone resorption area and pit number (Figure 7J, K). These data suggest that pharmacological and genetic targeting of PLD1 suppressed osteoclastogenesis and bone resorption.
We also added the following sentence in the legend (Line 311-314 of revised version):
“Representative TRAP staining images (H) and TRAP-positive cell numbers and TRAP activity (I) of the osteoclasts differentiated BMMs from PLD1+/+ and PLD1-/- mice. Representative of bone resorption images (J) and resorbed area and pit number (K) of the osteoclasts differentiated BMMs from PLD1+/+ and PLD1-/- mice."
Point 13: Based on data in Fig 7, PLD1 plays a critical role in osteoclastogenesis and bone resorption regardless of its Th17/treg balance. Any discussion on regulatory role of PLD1 in osteoclastogenesis and bone resorption?
Response 13: As you comment kindly, we described regulatory role of PLD1 in osteoclastogenesis and bone resorption in Discussion section (Line 366-377 of revised version): “Only a few studies have shown the effects of PLD in bone cells. PLD is involved in induction of the preresorptive cytokine IL-6, in osteoblasts [42]. PLD1 is also involved in the lung cancer-derived IL-8-induced osteoclstogenesis [43] and required for IL-15-mediated osteoclastogenesis in RASF [31]. These studies suggest that PLD can mediate the inflammatory response and induce bone resorption by stimulating osteoclast differentiation. Identification of PLD signaling pathway as the crucial regulator of osteoclastogenesis will lead to a better understanding of the mechanisms regulating osteoclastogenesis. At present, it is unknown whether PLD1 regulates osteoclastogenesis and bone resorption via modulation of Treg/Th17 balance. However, it is possible that regulation of Treg/Th17 cells by targeting of PLD1 in CIA mice might be involved in the suppression of osteoclastogenesis and bone loss. Future studies are needed to investigate the role of PLD1-mediated Treg/Th17 balance in osteoclastogenesis and bone resorption”.
Point 14: Line 365, what is booster material?
Response 14: Thank you for your kind comment. CII emulsified in incomplete Freund’s adjuvant was used as booster material. We added the following sentences in Materials and Methods (Line 390-393 of revised version): “Mice were injected intradermally at the base of the tail with 100 μg bovine type II collagen (CII) (Chondrex, Redmond, WA, USA) containing complete Freund’s adjuvant (Chondrex). Three weeks later, the mice were given boosters in the tail with 100 μg of CII, emulsified in incomplete Freund’s adjuvant (Chondrex)”.
Point 15: Line 366, provide PLD1 inhibitor treatment regimen.
Response 15: As you comment kindly, we provided the regimen of PLD1 inhibitor treatment in Materials and Methods (Line 393-394 of revised version): “For efficacy of PLD1 inhibitor, PLD1 inhibitor (5 mg/kg) was intraperitoneally injected every other day after arthritis onset on day 21”.
Point 16: Section 4.4. based on what described in this section, it seemed like to this reviewer that the authors measured total IgG1 and IgG2a not CII-specific autoantibodies in this study. Please clarify what the authors really measured in serum. Just IgG1 and IgG2a, or anti-bovine CII-specific IgG1 and IgG2a, or anti-mouse CII-specific IgG1 and IgG2a? Please provide detail procedures of this ELISA. What was coated in the ELISA plate?
Response 16: We apologize for not describing detail procedures of this ELISA. We coated murine type II collagen in the ELISA plate. We added the following sentences in Materials and Meyhods (Line 419-424 of revised version): “The amount of murine CII-specific autoantibodies in the serum were measured using the Mouse Anti-Type II Collagen IgG Subtype Antibody Assay Kit (Chondrex), according to the manufacturer’s instructions. In brief, microtiter plates were coated with 0.5 μg/well murine CII and incubated with serially diluted test sera. Bound IgG was detected by incubation with horseradish peroxidase (HRP)-conjugated anti-mouse IgG1 or IgG2a, and tetramethylbenzidine (TMB) substrate. Abservance (450 nm) was measured with an ELISA plate reader”.
Point 17: Section 4.7, How many events/samples were analyzed?
Response 17: As you pointed out, we added the number of analyzed samples (Line 435 of revised version): “Forty-two days after the primary immunization, the mouse spleens (n=6 per group) were dissected and washed twice with PBS for cell prepration”.
Point 18: S2, S3, & S5, indicate time of sampling.
Response 18: As you comment kindly, we indicated the time of sampling in S2, S3 and S5.
For Fig. S2, sampling time was indicated: “The sections of knee joint from the indicated mice with CIA at day 42, were analyzed by immunofluorescence staining using anti-TNF-α antibody and DAPI”.
For Fig. S3, sampling time was indicated: “(A) Representative images of spleen from the indicated mice (n=15, at day 45) and quantification of the spleen size (B) Representative images of H&E-stained sections and histopathological scores of liver and spleen from the indicated mice at day 45”.
For Fig. S5 (S7 in revised version), sampling time was indicatd: “The splenocytes were isolated from the indicated mice at day 45”. S5 was changed to S7 in revised version.
Point 19: S, S4, indicate sample size.
Response 19: As you comment kindly, we indicated the sample size in S4. (A) The knee joints from the indicated mice (n=6, at day 45) (B, C) RASF (n=3) were pretreated with PLD1 inhibitor (10 μM) and treated with TNF-α (10 ng/mL) or LPS (100 ng/mL) for 36 h.

Round 2
Reviewer 1 Report
Overall, the discussion should be re-written in a more structured way and better cited. The logic of the argumentation and style should be ameliorate. For example, line 358: "RA and CIA are T-cell-driven diseases and thus PLD1 might play a pivotal role in T cell activation."
detailed comments:
line 325: "reported to be associated with inflammation in RASF." The phrasing is vague. It is upregulation of inflammatory pathways?
line 343: Authors should insert citation.
line 219: It is very surprising/unlikely that RASF are producing IL17. The literature shows cross talk between IL-17 and RASF but not production of IL17 by RASF. same comment for IL10?
Figure S4: there is no C on the figure. legend does not match to the figure.
Figure 6: The gating strategy does not include live dead staining, CD4 and CD3 staining. Dead cells from in vitro culture could pick up antibody staining and bias the interpretation.
Author Response
Point 1: Overall, the discussion should be re-written in a more structured way and better cited. The logic of the argumentation and style should be ameliorate. For example, line 358: "RA and CIA are T-cell-driven diseases and thus PLD1 might play a pivotal role in T cell activation."
Response 1: Thank you very much for your valuable comments. As you suggest, we rewrote in a more structured way and ameliorated the logic of the argumentation.
For line 353 of revised version, "RA and CIA are T-cell-driven diseases and thus PLD1 might play a pivotal role in T cell activation." was changed to “RA and CIA are T-cell-driven diseases and the protection against the diseases is related to the inability of T effectors to be activated”.
Moreover, we better cited including citations of line 325.
To ameliorate the logic of the argumentation and style, we removed the following sentences in Discussion section: “PLD1 plays a pivotal role in IL-1β-induced activation of RASF and is a critical mediator of synovial inflammation [14]. Our findings indicate that PLD1 contributes to joint inflammation in RA. Previously, 1-butanol has been widely employed to identify PLD/PA-driven processes. However, 1-butanol does not always effectively reduce PA accumulation, and its use may result in PLD-independent, deleterious effects [39]. Several biological processes suppressed by 1-butanol are not affected by the small-molecule, PLD inhibitor [39]. It has also been suggested that reduced CTLA-4 expression by 1-butanol is a consequence of a nonspecific effect [19]”.
In addition, we rewrote the sentences of the previous version:
“PLD is involved in the induction of the preresorptive cytokine, IL-6, in osteoblasts [42]. PLD1 is also involved in lung cancer-derived IL-8-induced osteoclastogenesis [43] and is required for IL-15-mediated osteoclastogenesis in RASF [31]” were changed to “PLD is involved in the induction of the preresorptive cytokine, IL-6, in osteoblasts [40] and required for lung cancer-derived IL-8-induced osteoclastogenesis [41] and IL-15-mediated osteoclastogenesis in RASF [30]”.
Detailed comments:
Point 2: line 325: "reported to be associated with inflammation in RASF." The phrasing is vague. It is upregulation of inflammatory pathways?
Response 2: Thank you very much for your helpful comments. We rewrote the sentences of line 325 in Discussion section of revised version:
“Although PLD1 has been reported to be associated with inflammation in RASF [12,31,32]” was rewritten as follow: “Although PLD1 has been reported to be upregulated in IL-15-induced RASF, IL-1b-stimulated synoviocytes, and synovia from patients with RA [14, 30, 31]”.
Point 3: line 343: Authors should insert citation.
Response 3: Thank you for your comments. For the logic of argumentation, we removed the following sentences of previous version “The generation of pathogenic autoantibodies to CII by B cells might also be owing to an inefficient inflammatory response from the T-cell subset” was removed and thus did not insert citation. We described the following sentences in Discussion section: “Reduction of IgG2a levels by targeting PLD1 might ameliorate the severity of arthritis, suggesting a potential role of PLD1 in B cells”.
Point 4: line 219: It is very surprising/unlikely that RASF are producing IL17. The literature shows cross talk between IL-17 and RASF but not production of IL17 by RASF. same comment for IL10?
Response 4: Thank you very much for your valuable comments. We sincerely apologize for omitting exact information. We performed coculture experiment using RASF and peripheral blood mononuclear cells (PBMC) and measured production of the cytokine such as IL17 and IL10. In line 219 of revised version, we rewrote the sentence: “in RASF cocultured with peripheral blood mononuclear cells (PBMC) activated with anti-CD3 and CD28 antibodies (Figure S4B,C)”. In line 223, we changed RASF to RASF cocultured with PBMC. We described coculture assays in supplementary materials and method section, and legend of Figure S4B and C.
Point 5: Figure S4: there is no C on the figure. legend does not match to the figure.
Response 5: We sincerely apologize for omitting C on the legend of Figure S4. We added “C” in Figure S4.
Point 6: Figure 6: The gating strategy does not include live dead staining, CD4 and CD3 staining. Dead cells from in vitro culture could pick up antibody staining and bias the interpretation.
Response 6: As you kindly comment, we showed the gating strategy including live and dead staining and CD4 staining, in Figure S5. We added the following sentences in line 459 of revised version.
“For dead cell exclusion, the cells were resuspended in PBS containing 5 μl of 7-ADD (BD Biosciences)/1× 106 cells for 10 min, and washed twice. Following surface stain, the cells were incubated with anti-mouse CD16/CD32 (Fc blocker, Thermo Fisher Scientific), anti-mouse CD4 (Thermo Fisher Scientific), and anti-mouse CD25-APC antibodies (Thermo Fisher Scientific) for 20 min at 4oC. Intracellular staining was performed using a fixation/permeabilization kit (BD Biosciences), anti-mouse Foxp3-PE (Thermo Fisher Scientific), and anti-mouse IL-17-PE (Thermo Fisher Scientific), according to the manufacturer’s instructions”.
Reviewer 2 Report
The revised manuscript is very pleasant to read and the study was well conducted and conclusion is supported by solid data.
The following are very minor concerns:
- Line 134, “of TNFa positive cells in the spleen subjected to CIA, as …. (Figure S2)”. Figure S2 title and legend indicate that tissue source of this TNFa staining is synovial tissues not spleen. Clarify it.
- Line 143, “..population in splenic CD4+ T cells…” should be “in spleen”, it the parent gate is splenocytes.
- Line 146, “…among CD4+ T cells in the spleen..” should be “..among splenic CD4+ T cells”, if the parent gate is splenic CD4+ T cells.
- Figure 3 legend line 149, does n=6 per group mean 6 mice/group or 6 fields/mouse? Please state how many fields/mouse spleen and how many mice/group were analyzed.
- Figure 3, It looks like parent gate for A and B is splenocytes but parent gate for C is CD4+ T cells. To make things consistent, express them using the same parent gate for A, B, and C (either splenocytes or CD4+ T).
- Line 196-197 and line 390, “… day after the onset of arthritis on day 21”. On day 21, no animal developed arthritis. Change to “… day start on day 21 post immunization” or “…day after the booster injection on day 21”.
- Provide information on RASF and PBMC used in Figure S4. Human? Mouse?
- Figure S4, does PLD1 inhibitor affect cytokine production in RASF or PBMC (or any specific type of PBMC) or it affects cytokine production only in co-culture system?
- Figure 6 A and B, please indicate parent gate. Are they really % of splenocytes?
- Figure 6E legend (line 238-240) and Figure S8 legend are not clear (…for 72 h with non-Treg cells, with induced…). Clarify them.
- Figure 6E, were CD4+ Teff cells stimulated with anti-CD3+anti-CD28 to induce proliferation or are you observing spontaneous proliferation?
- Line 384, backcrossed how many generations?
- Line 393, missing citation.
- Line 398, Is it true that arthritis scoring was done by 10 independent observers? Not two?
Author Response
The revised manuscript is very pleasant to read and the study was well conducted and conclusion is supported by solid data.
The following are very minor concerns:
Point 1: Line 134, “of TNFa positive cells in the spleen subjected to CIA, as …. (Figure S2)”. Figure S2 title and legend indicate that tissue source of this TNFa staining is synovial tissues not spleen. Clarify it.
Response 1: Thank you very much for your valuable comments. As you suggest, we clarified it as “synovial tissues” in line 134.
Point 2: Line 143, “..population in splenic CD4+ T cells…” should be “in spleen”, it the parent gate is splenocytes.
Response 2: Thank you very much for your helpful comments. As you suggest, we rewrote it as “in spleen” in line 143.
Point 3: Line 146, “…among CD4+ T cells in the spleen..” should be “..among splenic CD4+ T cells”, if the parent gate is splenic CD4+ T cells.
Response 3: Thank you very much for your valuable comments. As you suggest, we rewrote it as “among splenic CD4+ T cells” in line 146.
Point 4: Figure 3 legend line 149, does n=6 per group mean 6 mice/group or 6 fields/mouse? Please state how many fields/mouse spleen and how many mice/group were analyzed.
Response 4: Thank you very much for your comments. We stated 3 fields/mice spleen and 6 mice/group in line 149 and 150.
Point 5: Figure 3, It looks like parent gate for A and B is splenocytes but parent gate for C is CD4+ T cells. To make things consistent, express them using the same parent gate for A, B, and C (either splenocytes or CD4+ T).
Response 5: Thank you very much for your valuable comments. As you suggest, we expressed the same parent gate for A, B, and C as “in total CD4+ T cells”.
Point 6: Line 196-197 and line 390, “… day after the onset of arthritis on day 21”. On day 21, no animal developed arthritis. Change to “… day start on day 21 post immunization” or “…day after the booster injection on day 21”.
Response 6: Thank you very much for your valuable comments. As you suggest, we rewrote it as “ . . . day start on day 21 post immunization” in line 196-197.
Point 7: Provide information on RASF and PBMC used in Figure S4. Human? Mouse?
Response 7: Thank you very much for your valuable comments. As you suggest, we provided information on RASF and PBMC as human source in the legend of Figure S4..
Point 8: Figure S4, does PLD1 inhibitor affect cytokine production in RASF or PBMC (or any specific type of PBMC) or it affects cytokine production only in co-culture system?
Response 8: Thank you very much for your valuable comments. As you suggest, we rewrote
Since coculture of RASF with activated PBMC mimic synovial environment, we investigated production of the cytokines in the coculture system. Thus, we did not perform cytokine production in RASF or PBMC.
Point 9: Figure 6 A and B, please indicate parent gate. Are they really % of splenocytes?
Response 9: Thank you very much for your helpful comments. As you suggest, we indicated parent gate as % of CD4+IL-17+ cells in splenocytes and % of CD4+Foxp3+ cells in splenocytes, respectively in Figure 6 A and B.
Point 10: Figure 6E legend (line 238-240) and Figure S8 legend are not clear (…for 72 h with non-Treg cells, with induced…). Clarify them.
Response 10: Thank you very much for your valuable comments. As you suggest, we rewrote and clarified it. The following sentence is shown in line 237-240 of Figure 6E and the legend of Figure S8 ,
“Freshly isolated naïve CD4+ T cells (naïve T cells), labeled with CFSE were cocultured in the presence of anti-mouse CD3ɛ and anti-mouse CD28 for 72 h with induced Treg cells (iTreg) generated in the presence or absence of the PLD1 inhibitor. Proliferation of naïve CD4+ T cells was analyzed by flow cytometry for CFSE dilution”.
“Fresh isolated naïve CD4+ T cells, labeled with CFSE were cocultured in the presence of anti-mouse CD3ɛ and anti-mouse CD28 for 72 h with induced Treg cells differentiated from PLD1+/+ and PLD1-/- T cells (n=6). Proliferation of naïve T cells was analyzed by flow cytometry for CFSE dilution”.
Point 11: Figure 6E, were CD4+ Teff cells stimulated with anti-CD3+anti-CD28 to induce proliferation or are you observing spontaneous proliferation?
Response 11: Thank you very much for your valuable comments. Naïve CD4+ T cells were stimulated with anti-CD3ɛ and anti-CD28 antibodies.
Point 12: Line 384, backcrossed how many generations?
Response 12: Thank you very much for your valuable comments. We backcrossed the mice more than at least 7 generation. We described it in line 384.
Point 13: Line 393, missing citation.
Response 13: Thank you very much for your valuable comments. As you suggest, we cited it as reference 22.
Point 14: Line 398, Is it true that arthritis scoring was done by 10 independent observers? Not two?
Response 14: Thank you very much for your comments. We apologize for a typo. We performed arthritis scoring by two independent observers. We rewrote it in line 397.
Round 3
Reviewer 1 Report
The authors made a great effort into reshaping the discussion. Overall, the flow and argumentation has been ameliorated supporting more clearly the presented data.
Author Response
Thank you very much for your positive commnets.